# Cell-type-specific Alzheimer's disease polygenic risk scores are associated with distinct disease processes in Alzheimer's disease

Hyun-Sik Yang ●[1,2,3,4] ✉, Ling Teng[1,4], Daniel Kang[2], Vilas Menon ●[5,6], Tian Ge[3,4,7,8], Hilary K. Finucane ●[3,4,9], Aaron P. Schultz[2,3], Michael Properzi[2], Hans-Ulrich Klein ●[5,6], Lori B. Chibnik ●[2,4,10], Julie A. Schneider ●[11], David A. Bennett[11], Timothy J. Hohman ●[12], Richard P. Mayeux[6], Keith A. Johnson[1,2,3,13], Philip L. De Jager ●[5,6] & Reisa A. Sperling ●[1,2,3]

Many of the Alzheimer's disease (AD) risk genes are specifically expressed in microglia and astrocytes, but how and when the genetic risk localizing to these cell types contributes to AD pathophysiology remains unclear. Here, we derive cell-type-specific AD polygenic risk scores (ADPRS) from two extensively characterized datasets and uncover the impact of cell-type-specific genetic risk on AD endophenotypes. In an autopsy dataset spanning all stages of AD ($n = 1457$), the astrocytic ADPRS affected diffuse and neuritic plaques (amyloid-β), while microglial ADPRS affected neuritic plaques, microglial activation, neurofibrillary tangles (tau), and cognitive decline. In an independent neuroimaging dataset of cognitively unimpaired elderly ($n = 2921$), astrocytic ADPRS was associated with amyloid-β, and microglial ADPRS was associated with amyloid-β and tau, connecting cell-type-specific genetic risk with AD pathology even before symptom onset. Together, our study provides human genetic evidence implicating multiple glial cell types in AD pathophysiology, starting from the preclinical stage.

Alzheimer's disease (AD) is the most common cause of dementia and is among the leading causes of death, but clinically effective disease-modifying intervention has been challenging[1]. A major barrier to AD drug discovery is the complex pathophysiology driven by multiple cell types interacting with amyloid-β (Aβ) and tau proteinopathies[1–4]. Large genome-wide association studies (GWAS) of AD dementia have identified dozens of candidate causal genes, many highly expressed in microglia and astrocytes[5–7]. Further, numerous sub-threshold genetic associations are also enriched in microglial genes[8–13]. Microglia are resident immune cells of the central nervous system (CNS), implicated in Aβ clearance, Aβ-related neuroinflammation, and Aβ- and tau-related neurodegeneration[1,2]. Astrocytes are the hub of lipoprotein and cholesterol metabolism in the CNS, a process closely related to Aβ metabolism, and are also implicated in neurodegeneration[2,14]. Both

[1]Center for Alzheimer Research and Treatment, Department of Neurology, Brigham and Women's Hospital, Boston, MA, USA. [2]Department of Neurology, Massachusetts General Hospital, Boston, MA, USA. [3]Harvard Medical School, Boston, MA, USA. [4]Broad Institute of MIT and Harvard, Cambridge, MA, USA. [5]Center for Translational & Computational Neuroimmunology, Department of Neurology, Columbia University Irving Medical Center, New York, NY, USA. [6]Department of Neurology and the Taub Institute for the Study of Alzheimer's Disease and the Aging Brain, Columbia University Irving Medical Center, New York, NY, USA. [7]Psychiatric and Neurodevelopmental Genetics Unit, Center for Genomic Medicine, Massachusetts General Hospital, Boston, MA, USA. [8]Center for Precision Psychiatry, Department of Psychiatry, Massachusetts General Hospital, Boston, MA, USA. [9]Department of Medicine, Massachusetts General Hospital, Boston, MA, USA. [10]Department of Epidemiology, Harvard T.H. Chan School of Public Health, Boston, MA, USA. [11]Rush Alzheimer's Disease Center, Rush University Medical Center, Chicago, IL, USA. [12]Department of Neurology, Vanderbilt University Medical Center, Nashville, TN, USA. [13]Department of Radiology, Massachusetts General Hospital, Boston, MA, USA. ✉e-mail: hyang18@bwh.harvard.edu

activated microglia and astrocytes are found proximate to neuritic Aβ plaques[1,2,15], a pathologic hallmark of AD. With strong support from human genetics and accumulating experimental evidence, microglia, and astrocytes are emerging as promising cellular targets for potential disease-modifying interventions in AD.

However, how and when the AD genetic risk localizing to these cell types contributes to distinct processes in AD remains unclear, making it very difficult to design clinical trials that can precisely target the right cellular program at the right disease stage. The progression of AD is driven by multiple related yet distinct disease processes that gradually progress over more than two decades, leading to a substantial clinical-pathological heterogeneity[3]. There is significant variability in the rate of in vivo tau accumulation in individuals with similar Aβ burden[16], and the rate of cognitive decline is highly variable even in the setting of similar Aβ and tau burden (cognitive resilience)[17–19], which have been attributed to the differential cellular responses to neuropathologic insults[4,18,20–23]. On the other hand, well-powered case-control AD dementia GWAS focus on dementia diagnosis (the final outcome of AD progression) and does not account for individual-level AD endophenotype (e.g., Aβ, tau, cognitive decline) variability, lacking the resolution to localize the cell-type-specific genetic association to a specific AD endophenotype ("how") at a specific stage of disease progression ("when"). AD endophenotype GWAS in deeply characterized individuals is a promising approach to fill this gap, but despite recent growth in sample size[21,24–26], these studies are not yet powered for robust cell-type-specific heritability analyses.

Therefore, we need an approach to combine well-powered AD dementia GWAS results with deeply characterized individual-level data in a cell-type-specific manner. Previous studies have identified direct associations of several AD dementia GWAS variants with key AD endophenotypes such as Aβ plaques[27–29], tau tangles[30], and cognitive decline[29,30]. Yet, most AD dementia GWAS loci have small effect sizes, and their association with the AD endophenotypes cannot be robustly examined with moderate sample sizes of well-characterized datasets. Aggregating genetic effects with AD polygenic risk scores (ADPRS) can enable robust detection of overall genetic effects on AD endophenotypes[31–33], but conventional ADPRS lacks cellular specificity. In this context, an emerging alternative is a gene-set-based PRS approach[13,34–38] to capture cell-type-specific AD genetic risk profiles from each individual. Previous studies have applied gene-set-based ADPRS to predict AD dementia in a pathway- or cell-type-specific manner[13,37,38]. Nonetheless, the relationships between cell-type-specific AD genetic risk and distinct AD endophenotypes (e.g., Aβ, tau, cognitive decline) remain largely unknown.

Here, we derive single nucleus RNA sequencing (snRNA-seq)-guided cell-type-specific ADPRS from two extensively characterized datasets and clarify how and when AD genetic risk localizing to each major brain cell type contributes to distinct disease processes in AD. We first leverage detailed post-mortem quantitative neuropathology data from two community-based cohorts that span the full pathologic and clinical disease severity spectrum. We observe specific associations of astrocytic (Ast) and microglial (Mic) ADPRS with distinct AD endophenotypes and perform causal modeling analyses. Then, we focus on the preclinical (asymptomatic) stage of AD and examine the association between cell-type-specific ADPRS and neuroimaging AD biomarkers in a clinical trial screening dataset, replicating our key findings and establishing the early role of astrocytic and microglial genetic risk in AD pathogenesis.

## Results

### Study participants

Our study participants are from two independent datasets. First, we examined the impact of cell-type-specific ADPRS on the longitudinal cognitive and post-mortem neuropathology data from the Religious Orders Study and the Rush Memory and Aging Project (ROSMAP) ($n = 1457$, mean age 89.7 ± 6.5, 67% female, 69% with elevated Aβ, 45% with dementia; Table 1). ROS and MAP are community-based cohorts with annual cognitive exams and comprehensive postmortem neuropathologic evaluation, and the full spectrum of pathologic and clinical stages of AD are represented[39]. Second, we analyzed the genetic and phenotypic data from the pre-randomization (screening) phase of the Anti-Amyloid Treatment in Asymptomatic Alzheimer's (A4) study[40], a secondary AD prevention trial. The A4 screening dataset consists of CU older adults with florbetapir positron emission tomography (PET) evaluation ($n = 2921$, mean age 71.4 ± 4.8, 60% female, 30% with elevated Aβ; Table 1), enabling us to test whether cell-type-specific ADPRS also impacts the in vivo AD endophenotypes in the earliest stages of AD.

### Derivation of cell-type-specific ADPRS

We adapted and combined previously described approaches[10,11,13,34–38,41,42] to derive cell-type-specific ADPRS (Fig. 1a). We first used a Bayesian regression approach (PRS-CS)[41] to perform effect size shrinkage on the base AD GWAS summary statistics (Bellenguez et al., stage I)[5]. PRS-CS assigns posterior effect sizes for each genetic variant based on the GWAS summary statistics and linkage disequilibrium (LD) structure and does not prune the linked SNPs. Thus, PRS-CS retains more SNPs and reduces information loss, compared to the widely used LD clumping methods that only retain one lead SNP per LD block (Supplementary Table 1). Further, PRS-CS only uses the GWAS summary statistics for optimization, avoiding overfitting to the target datasets (see Methods). Then, we used a previously published neocortical snRNA-seq dataset from 24 controls[43] (ROSMAP participants with no or very little pathology; no participant overlap with our study) and derived cell-type-specific gene lists from six major brain cell types: excitatory neurons (Ex), inhibitory neurons (In), astrocytes (Ast), microglia (Mic), oligodendrocytes (Oli), and oligodendrocyte precursor cells (Opc). After removing the *APOE* region (*APOE* ± 1 MB), we defined a cell-type-specific gene list by selecting genes within the top 10% of expression specificity[10,11] for each cell type (i.e., top 1343 genes). Only a minor proportion of cell-type-specific genes were specific to two or more brain cell types (Fig. 1b), and we allowed this overlap as some genes may have important roles in multiple (but not all) cell types. Each cell-type-specific ADPRS was computed from ROSMAP and A4, using 30 kb margins upstream and downstream of cell-type-specific genes. The cell-type-specific ADPRSs included 7.6–10.4% of all examined variants (Supplementary Table 1), and they were orthogonal to each other (R$^2$ < 0.1), except for the Mic- and Oli-ADPRS pair with a strong positive correlation (R$^2$ = 0.31 in ROSMAP, R$^2$ = 0.26 in A4) (Fig. 1c, d). Much of the shared variance between Mic- and Oli-ADPRSs

## Table 1 | Study participant characteristics

|  | ROSMAP (*n* = 1457) | A4 (*n* = 2921) |
|---|---|---|
| Mean Age, years (SD) | 89.7 (6.5) | 71.4 (4.8) |
| Female (%) | 973 (67) | 1740 (60) |
| Mean Education, years (SD) | 16.3 (3.6) | 16.7 (2.7)[a] |
| *APOE* ε4 carrier (%) | 376 (26) | 1038 (36) |
| Mean Florbetapir, cortical SUVR (SD) | NA | 1.10 (0.19) |
| Elevated Aβ (%) | 1008 (69) | 890 (30)[b] |
| Pathological diagnosis of AD | 954 (65) | NA |
| Median MMSE (IQR) | 24 (13) | 29 (2) |
| All-cause dementia (%) | 661 (45) | 0 (0) |
| AD dementia (%) | 539 (37) | 0 (0) |

*AD* dementia, AD with dementia, *APOE* apolipoprotein E, *IQR* interquartile range, *MMSE* Mini-Mental State Examination, *SD* standard deviation, *SUVR* standardized uptake value ratio (whole cerebellar reference).
[a] *n* = 2919 with data.
[b] *n* = 2920 with data.

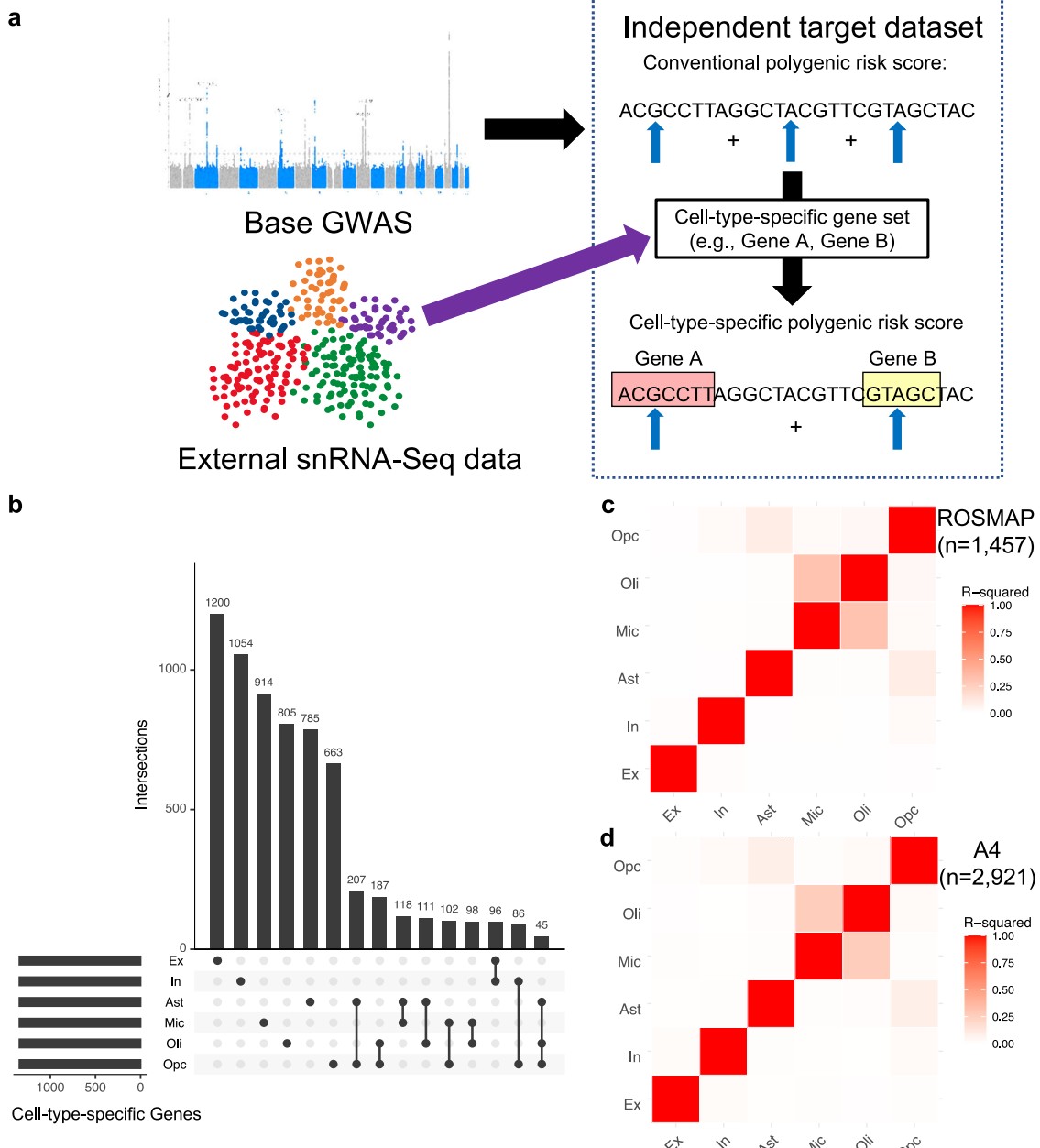

**Fig. 1 | Cell-type-specific Alzheimer's disease polygenic risk scores (ADPRS). a** A schematic of cell-type-specific PRS derivation. **b** An UpSetR plot of cell-type-specific gene sets used to define cell-type-specific ADPRS. Each cell-type-specific gene set includes genes within the top 10% of cell-type specificity ($n = 1343$). Each row of the matrix represents each cell-type-specific gene set, and each column of the matrix represents an intersection of one or more sets. Gene sets in each intersection were indicated by filled black circles connected by a black vertical line. The vertical bar graph on the top shows the number of genes in each intersection.

The 15 most frequent intersections were visualized. **c, d** Correlation matrix among cell-type-specific ADPRS (**c** ROSMAP; **d** A4). Pearson's correlation coefficient was positive for all pairs, and the square of Pearson's correlation coefficient ($R^2$) between pairs of cell types was visualized. Source data are provided as a Source Data file. Ast astrocytes, Ex excitatory neurons, In inhibitory neurons, Mic microglia, Oli oligodendrocytes, Opc oligodendroglial progenitor cells, snRNA-Seq single nucleus RNA-sequencing.

remained even when the PRSs were derived without genomic margin ($R^2 = 0.21$ in ROSMAP, $R^2 = 0.18$ in A4), indicating that the correlation between the two is likely due to overlapping genes rather than overlapping genomic margins. Mic-specific and Oli-specific gene sets shared 136 genes (10.1% of each set), which include known AD risk genes[5], such as *ADAM10*, *BIN1*, *CR1*, and *PICALM*.

### Cell-type-specific ADPRSs were associated with distinct AD endophenotypes in ROSMAP

Then, we tested the association of cell-type-specific ADPRSs with seven AD endophenotypes in ROSMAP: autopsy-confirmed AD

dementia, immunohistochemistry (IHC)-assessed Aβ and paired helical filament tau (PHFtau), Bielschowsky silver stain-assessed diffuse plaque (DP), neuritic plaque (NP), and neurofibrillary tangle (NFT), and longitudinally assessed cognitive decline (Supplementary Table 2). IHC enables a molecularly specific and quantitative assessment of Aβ and tau pathology, while the silver stain allows a separate assessment of specific morphological subtypes of amyloid plaques (DP vs. NP). The ADPRS derived from the entire autosomal genome, excluding the *APOE* region (All-ADPRS), was associated with all endophenotypes except for DP (Fig. 2 and Supplementary Tables 3–9).

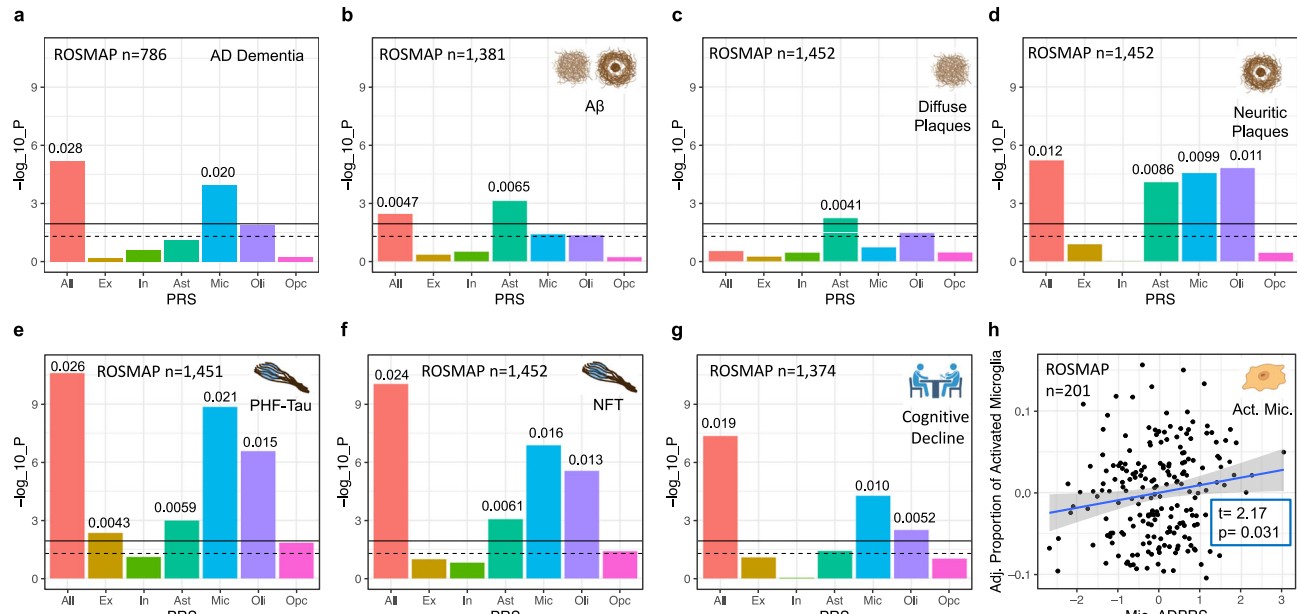

**Fig. 2 | Association of cell-type-specific AD polygenic risk scores in ROSMAP.**
**a** Association of cell-type-specific ADPRSs with the odds of AD with dementia (case: $n = 538$, control: $n = 248$). **b** Association of cell-type-specific ADPRSs with overall Aβ burden. **c** Association of cell-type-specific ADPRSs with overall diffuse plaque (DP) burden. **d** Association of cell-type-specific ADPRSs with overall neuritic plaque (NP) burden. **e** Association of cell-type-specific ADPRSs with overall paired-helical-filament tau (PHFtau) burden. **f** Association of cell-type-specific ADPRSs with overall neurofibrillary tangle (NFT) burden. **g** Association of cell-type-specific ADPRSs with cognitive decline (the slope of annual change in antemortem measures of global cognitive composite). For **a**–**g**, the y-axis indicates -log₁₀(p-value) of each association, the black solid horizontal line indicates the p-value corresponding to multiple comparisons-corrected statistical significance (FDR = 0.025), and the black dashed horizontal line indicates $p = 0.05$. The p-values (two-sided, unadjusted) are from regression models (**a** logistic regression, **b**–**g** linear regression)

adjusting for *APOE* ε4, *APOE* ε2, age at death, sex, genotyping platform, years of education (only for **a** and **g**), and the first three genotyping principal components. The effect size of each statistically significant PRS-trait association was quantified with ΔR² (difference of Nagelkerke's R² (**a**) or adjusted R² (**b**–**g**) between the linear models with and without the given PRS term; **b**–**g**) and indicated above the bar graph. Also, see Supplementary Tables 3–9 for further details. Source data are provided as a Source Data file. **h** Association of Mic-ADPRS (x-axis) and the proportion of activated microglia (PAM, y-axis). On y-axis, residual PAM values adjusting for covariates (*APOE* ε4, *APOE* ε2, age at death, sex, genotyping platform, and the first three genotyping principal components) were shown. T-statistics and p-values from linear regression (adjusting for the covariates) are shown. Created with Biorender.com. Act. Mic. activated microglia, All full autosomal genome, Ast astrocytes, Ex excitatory neurons, In inhibitory neurons, Mic microglia, Oli oligodendrocytes, Opc oligodendroglial progenitor cells.

Mic-ADPRS was the only cell-type-specific ADPRS significantly associated with increased odds of autopsy-confirmed AD dementia (Fig. 2a and Supplementary Table 3), consistent with the previous studies focusing on cell-type-specific heritability of AD dementia[5–13].

Ast-ADPRS was the only cell-type-specific ADPRS significantly associated with all three tested measures of post-mortem fibrillar Aβ burden: IHC-assessed Aβ and silver-stain assessed DP and NP (Fig. 2b–d and Supplementary Tables 4–6). Oli-, and Mic-ADPRSs were more specifically associated with the NP burden (Fig. 2d and Supplementary Table 6). DP is an amorphous aggregation of Aβ with minimal cellular reaction, while NP contains a dense core with surrounding neuroglial reaction including dystrophic neurites, activated microglia, and reactive astrocytes[44,45]. Although DP and NP burden are highly correlated (Pearson's $r = 0.69$ in our study), only NP was associated with multiple glial ADPRS, supporting that the observed cell-type-specific ADPRS–trait association was not driven by the correlation between AD endophenotypes. Thus, our finding suggests that astrocytic genetic programs contribute to Aβ accumulation starting from the early stages of fibrillar Aβ formation, perhaps by shifting the balance between Aβ production and clearance: this program might have the greatest impact on the overall fibrillar Aβ level. On the other hand, microglial and oligodendrocytic genetic programs may primarily contribute to cellular reaction to Aβ leading to NP formation[46–48].

Multiple cell-type-specific ADPRSs (Mic-, Oli-, Ast-, and Ex-ADPRSs) were associated with IHC-assessed PHFtau burden, and the strongest association was observed with Mic-ADPRS (Fig. 2e and Supplementary Table 7). Oli-, Ast-, and Ex-ADPRS remained nominally associated ($p < 0.05$) with PHFtau even after adjusting for Mic-ADPRS

(Supplementary Table 10), and Oli-, Ast-, and Ex-ADPRS calculated after excluding genes overlapping with microglia were all associated with PHFtau (Supplementary Table 11). Mic-, Oli-, and Ast-ADPRS were also associated with silver stain-assessed NFT burden, while Ex-ADPRS −NFT association did not reach statistical significance (Fig. 2f and Supplementary Table 8).

Mic- and Oli-ADPRS were significantly associated with cognitive decline (Fig. 2g and Supplementary Table 9), but Oli-ADPRS was no longer associated with cognitive decline after adjusting for Mic-ADPRS ($p = 0.41$). We note that cognitive decline in older adults is a complex phenomenon: less than 50% of the variability can be explained even after considering multiple pathologies and other known contributors[18,49]. This complexity might have undermined statistical power to detect the weaker associations, leaving Mic-ADPRS as the only cell-type-specific ADPRS independently associated with cognitive decline and dementia after accounting for multiple testing corrections.

In a subset of ROSMAP participants with morphological assessments of microglial activation ($n = 201$ MAP participants, demographics summarized in Supplementary Table 12), we explored whether Mic-ADPRS is associated with the proportion of activated microglia (PAM[50]). Histologically characterized microglial activation from the neocortex has a strong association with AD pathology, but its associations with known AD risk variants, including *APOE* ε4 ($p = 0.85$ in our study), were not significant at the single variant level[50]. Mic-ADPRS was associated with an increased PAM (Fig. 2h; beta = $9.2 \times 10^{-3}$, 95% CI $8.4 \times 10^{-4}$ to 0.017, $p = 0.031$). Thus, our cell-type-specific ADPRS showed that a higher microglial AD genetic risk may lead to

dysfunctional microglial activation, a relationship that was not apparent at the single AD GWAS variant level[50].

All 15 significant associations between cell-type-specific ADPRSs and AD endophenotypes remained similar even when we used different genomic margins for PRS (results from genes ±10 kb or ±100 kb; Supplementary Table 13) or adjusted for the higher number of genotype principal components (Supplementary Table 14). We also benchmarked our results against the PRSet[13], a previously published gene-set-based PRS approach that uses an LD clumping approach. Our PRS-CS-based cell-type-specific ADPRSs explained greater variances ($\Delta R^2$) than PRSet scores in 14 out of 15 endophenotype-PRS associations (Supplementary Table 15 and Supplementary Fig. 1; 10 with empiric bootstrap $p < 0.05$). Thus, our PRS-CS-based approach−that retains more cell-type-specific variants (Supplementary Table 1) and local genomic information−showed a superior statistical power than the existing LD-clumping-based approach. None of the observed trait−cell-type-specific ADPRS associations showed significant statistical interactions with age, sex, or *APOE* ε4 dosage (=no significant effect moderation).

To summarize, multiple cell-type-specific ADPRS were associated with AD endophenotypes in ROSMAP, consistent with the view that multiple cell types might contribute to AD pathophysiology[1–4]. We performed causal modeling analyses, as detailed in the next section, to further clarify the relationship between cell-type-specific ADPRSs and AD endophenotypes.

### Causal modeling analyses mapped Mic- and Ast-ADPRS to distinct events in the AD pathophysiologic cascade

We performed causal modeling to map the contribution of each cell-type-specific ADPRS to the sequence of events in AD pathophysiology, focusing on Mic- and Ast-ADPRS that showed significant associations with multiple AD endophenotypes. Although Oli-ADPRS was also strongly associated with NP and tau, we had to exclude it from this modeling approach, given the difficulty in statistically separating its effect from the colinear Mic-ADPRS. Here, we leveraged genetic risk scores that are assigned randomly at conception and are not subject to reverse causation[51]. We also used the postulated sequence of AD progression as the prior for our model: Aβ accumulation starts as DP, which evolves into NP with surrounding gliosis, triggering tau NFT formation and cognitive decline[4,44].

We first performed causal mediation analyses to distinguish direct and mediated effects among cell-type-specific ADPRSs and AD endophenotypes (Fig. 3a−d and Supplementary Table 16). Ast-ADPRS had a direct effect on NP not fully mediated by DP (Fig. 3a), while the direct Ast-ADPRS − NFT association was no longer significant after considering the NP-mediated effect (Fig. 3b). On the other hand, there were significant direct effects of Mic-ADPRS on NFT and cognitive decline even after accounting for the upstream processes (Fig. 3c, d).

Synthesizing these results, we constructed a structural equation model (SEM) from $n = 1392$ ROSMAP participants with no missing data (Fig. 3e). This SEM has an excellent model fit and highlights the distinct contribution of each cell-type-specific ADPRS: Ast-ADPRS affects AD pathophysiology mainly through its effect on Aβ (diffuse and neuritic plaques), while Mic-ADPRS has a broader impact on multiple core pathological and clinical endophenotypes of AD (NP, NFT, and cognitive decline). Moreover, Mic-ADPRS influenced cognitive decline above and beyond AD pathology, suggesting the role of microglia in cognitive resilience. We acknowledge that this model derived using post-mortem cross-sectional neuropathology data cannot prove a causal relationship. Still, our approach provides a plausible model based on human genetics that can inform future mechanistic studies.

### Cell-type-specific ADPRSs were associated with in vivo AD biomarkers in CU older adults

Then, we used in vivo neuroimaging biomarker data from CU older adults in the A4 screening data and assessed the role of cell-type-specific AD genetic risk in preclinical AD. We tested four AD endophenotypes in A4 (Aβ PET, tau PET, hippocampal volume [HV; a marker of neurodegeneration], and Preclinical Alzheimer Cognitive Composite [PACC[52]; Supplementary Table 17). All-ADPRS was associated with all tested AD endophenotypes, while cell-type-specific ADPRSs showed distinct association patterns (Fig. 4).

Ex-, Ast-, Mic-, and Oli-ADPRS were significantly associated with in vivo Aβ (FDR < 0.025; Fig. 4a and Supplementary Table 18). Ex- and Ast-ADPRS remained nominally associated with Aβ after adjusting for Mic-ADPRS or excluding genes overlapping with ADPRS, while Oli-ADPRS− Aβ association was no longer present (Supplementary Tables 19, 20). These results indicate that the genetic architecture of Aβ measured by PET is more similar to NP (Fig. 2d, Ast/Mic/Oli-ADPRS associations) rather than DP (Fig. 2c, only Ast-ADPRS association); this is likely because the PET radiotracers for Aβ have a greater affinity to NP than DP[53,54]. A larger sample size in A4 enabled us to detect the additional Ex-ADPRS −Aβ association.

In a smaller subset with tau PET data ($n = 302$; demographics summarized in Supplementary Table 21), only Mic-ADPRS was significantly associated (FDR < 0.025) with tau (temporal lobe composite) (Fig. 4b and Supplementary Table 22). This differs from multiple cell-type-specific ADPRS associations with PHFtau and NFT in ROSMAP. Interestingly, in a subset of ROSMAP participants who were CU ($n = 454$), only Mic-ADPRS was associated with PHFtau (beta = 0.16, $p = 1.0 \times 10^{-4}$), while Oli-, Ast-, and Ex-ADPRS were not ($p > 0.05$; Supplementary Table 23). Thus, despite important differences in phenotype measurement, results from ROSMAP and A4 suggest a coherent biology: microglia may exacerbate tau pathology starting from the preclinical stage of AD, while other cell types may contribute to tau pathology later in the symptomatic disease stages.

On the other hand, there was a limited cell-type-specific ADPRS association with neurodegeneration (HV) or cognition (PACC) in A4. In a subset with structural MRI data ($n = 1266$; demographics summarized in Supplementary Table 24), HV was not associated with any cell-type-specific ADPRSs (Fig. 4c and Supplementary Table 25). PACC, a sensitive cognitive composite optimized to detect early Aβ-related cognitive decline[52], was only associated with Ast-ADPRS (Fig. 4d and Supplementary Table 26), and this association remained similar even after adjusting for PET-measured Aβ (beta = −0.10, $p = 0.015$). The A4 study is likely underpowered to detect the impact of cell-type-specific AD genetic risk on neurodegeneration or cognitive impairment because all participants in the A4 screening dataset were CU without extensive AD-related neurodegeneration or cognitive decline. Nonetheless, the Aβ-independent association between Ast-ADPRS and PACC hints at a possible effect of Ast-ADPRS on early cognitive decline above and beyond AD pathology.

All six significant associations between cell-type-specific ADPRS and AD endophenotypes in A4 were robust to the size of genomic margins (genes ±10 kb or ±100 kb; Supplementary Table 27) or the number of genotype PCs adjusted for (Supplementary Table 28). Our PRS-CS-based cell-type-specific ADPRSs explained greater variances ($\Delta R^2$) than all corresponding PRSet scores (Supplementary Table 29, Supplementary Fig. 2; 3 out of 6 with empiric bootstrap $p < 0.05$). None of the observed trait−cell-type-specific ADPRS associations showed significant statistical interactions with age, sex, or *APOE* ε4 dosage (=no significant effect moderation).

## Discussion

We derived cell-type-specific ADPRSs in two independent and well-characterized datasets to show that AD genetic risk localizing to different neuroglial cell types makes distinct contributions to AD pathophysiology. Our findings provide human genetics evidence to support the disease model where astrocytes play an important early role in Aβ clearance before plaque maturation, while microglia are primarily involved in later phases of Aβ plaque maturation (i.e., NP

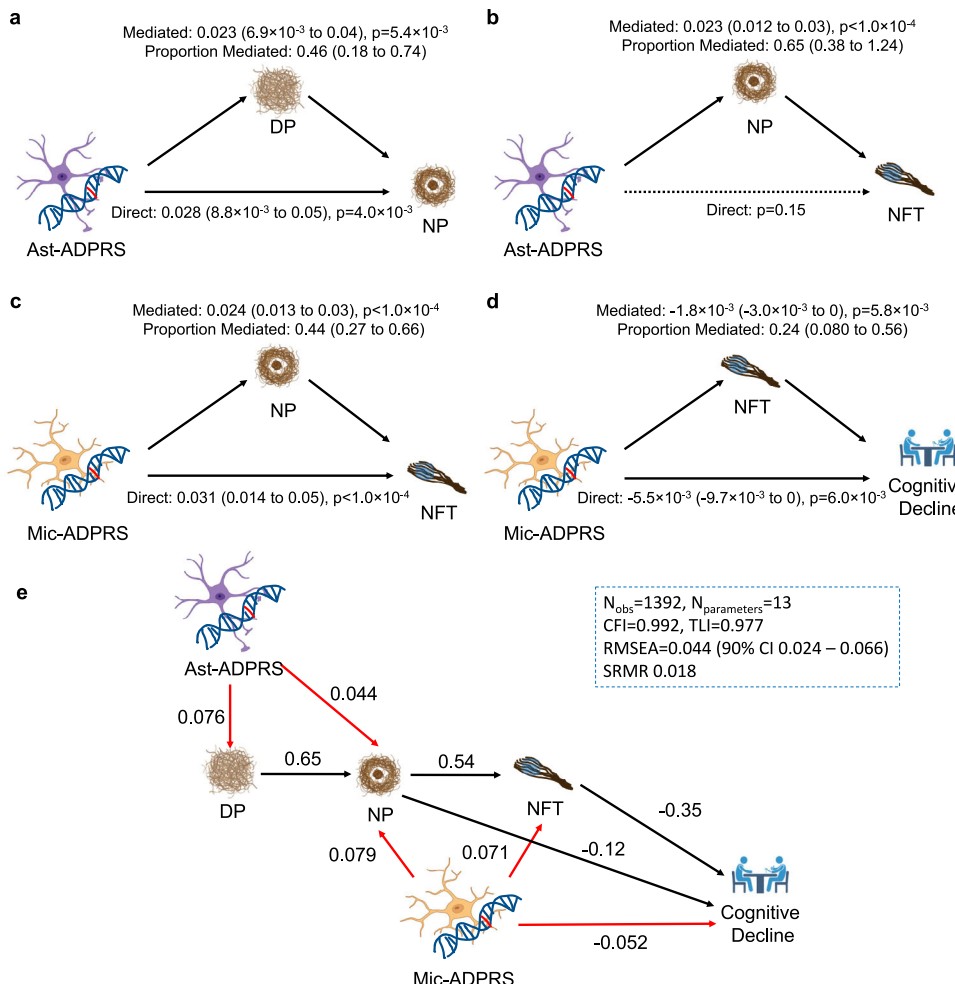

**Fig. 3 | Causal mediation analyses and structural equation modeling of cell-type-specific ADPRS and AD endophenotypes. a** DP partially mediates Ast-ADPRS−NP association ($n = 1452$). **b** NP mediates most of the Ast-ADPRS−NFT association ($n = 1474$), and the direct effect of Ast-ADPRS on NFT is not significant. **c** NP partially mediates Mic-ADPRS−tau association ($n = 1474$). **d** NFT partially mediates Mic-ADPRS−cognitive decline association ($n = 1392$). The model included NP burden as a covariate. All models in **a**−**d** are linear models adjusted for age, sex, education (for cognitive decline slope), *APOE* ε2 count, *APOE* ε4 count, genotype batch, and first three genotype principal components, and non-parametric bootstrapping ($n = 10{,}000$ iterations) were used to derive empiric two-sided $p$-values and 95% confidence intervals for the average causal mediated effect, average direct effect, and proportion mediated. Also, see Supplementary Table 16 for further details. **e** Structural equation modeling (SEM) shows a probable causal relationship

between cell-type-specific ADPRS and AD endophenotypes. Black solid arrows indicate phenotype-phenotype associations and red solid arrows indicate genotype-phenotype associations. All depicted associations were nominally significant ($p < 0.05$). Numbers adjacent to each arrow indicate completely standardized solutions (relative strength of the effect). Model fit metrics indicate an excellent model fit. All linear associations in this SEM are adjusted for age, sex, education (for cognitive decline slope), *APOE* ε2 count, *APOE* ε4 count, genotype batch, and first three genotype principal components. Created with Biorender.com. CFI comparative fit index, DP diffuse plaque, $N_{obs}$ number of observations (participants), $N_{parameter}$ number of model parameters, NP neuritic plaque, n.s. not significant, RMSEA root mean square error of approximation, SRMR standardized root mean square residual, TLI Tucker Lewis Index.

formation) and abnormal tau accumulation. Lipoprotein and cholesterol metabolism, primarily driven by astrocytes in the CNS, are implicated in Aβ metabolism[2]. Further, Aβ clearance occurs through the blood-brain barrier (BBB) and perivascular circulation, and astrocytes are among the main constituents of the BBB[2]. Thus, AD genetic risk localizing to astrocyte-specific genes may collectively undermine the Aβ metabolism and perivascular Aβ clearance, which leads to initial parenchymal fibrillar Aβ accumulation (DP). Then, dysfunctional microglial activation—which the aggregate microglial AD genetic risk may drive—and subsequent ineffective fibrillar Aβ removal may lead to NP formation, with additional contributions from astrocytes. On the other hand, tau pathology accumulation is likely to be initially driven by microglia in the preclinical stage, with later contributions of astrocytes, oligodendrocytes, and cell-autonomous actions of excitatory neurons. Interestingly, a recent spatial transcriptomics study from mice[15] showed that Aβ plaques are surrounded immediately by

microglia and more distantly by astrocytes, while hyperphosphorylated tau affects excitatory neurons in an environment enriched with oligodendrocytes, providing a landscape coherent with what our human genetics study suggests.

It is important to note that each cell-type-specific ADPRS explained 3% or less of the variance in each AD endophenotype. Thus, the current cell-type-specific ADPRS is unlikely to be a useful stand-alone tool for clinical trial screening or disease risk stratification. Nonetheless, our study demonstrates that cell-type-specific PRS can be used to gain deeper pathophysiologic insights from well-characterized cohorts and guide future mechanistic and clinical-translational studies. For example, cell-type-specific PRS could be leveraged for a genetically guided sampling of induced pluripotent stem cell (iPSC) lines for specific cell type differentiation, or it can be used for cell-type-specific pharmacogenomic studies of anti-Aβ immunotherapies. Further, our study provides genetic support to consider in vivo Aβ and tau PET−

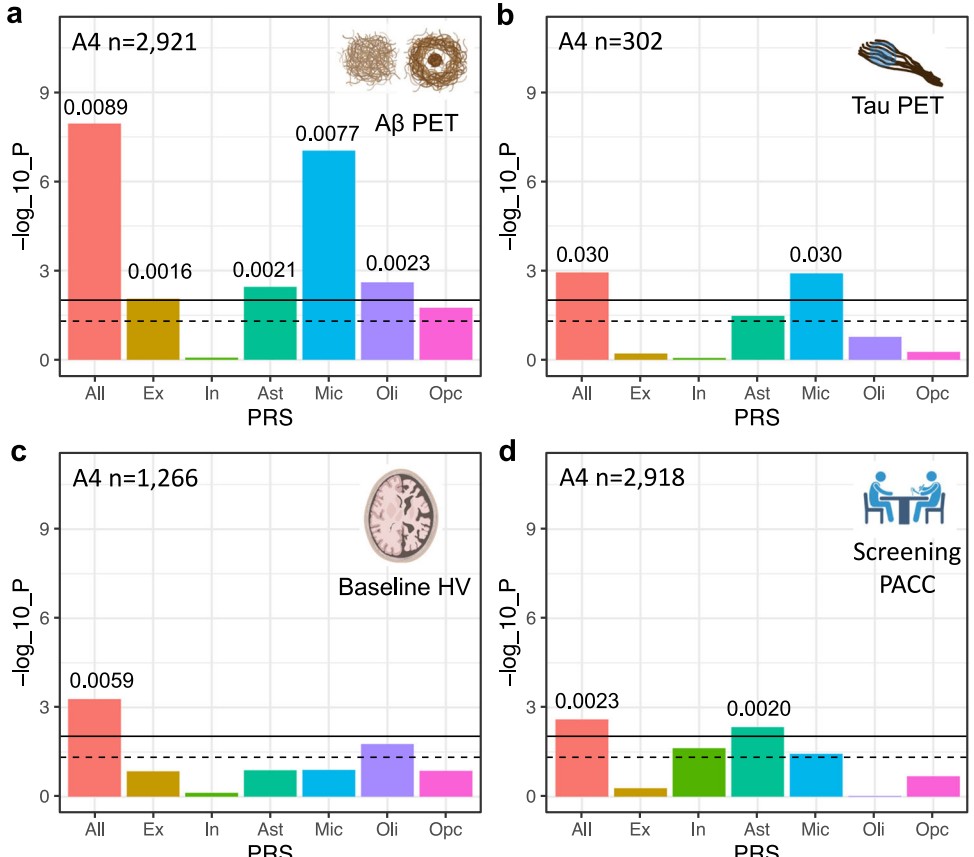

**Fig. 4 | Association of cell-type-specific AD polygenic risk scores in A4.**
**a** Association of cell-type-specific ADPRSs with cortical Aβ (florbetapir PET cortical composite SUVR). **b** Association of cell-type-specific ADPRSs with temporal lobe tau (flortaucipir PET temporal lobe composite SUVR). **c** Association of cell-type-specific ADPRSs with hippocampal volume (HV). **d** Association of cell-type-specific ADPRSs with screening Preclinical Alzheimer Cognitive Composite (PACC). The *y*-axis indicates -log₁₀(*p*-value) of each association. The black solid horizontal line indicates the *p*-value corresponding to statistical significance (FDR = 0.025), and the black dashed horizontal line indicates $p = 0.05$. The *p*-values (two-sided,

unadjusted) are from linear regression models adjusting for *APOE* ε4, *APOE* ε2, age, sex, intracranial volume (only for **c**), years of education (only for **d**), and the first three genotyping principal components. Effect size of each statistically significant PRS-trait association was quantified with ΔR² (difference of adjusted R2 between the linear models with and without the given PRS term) and indicated above the bar graph. Also, see Supplementary Tables 18, 22, 25, and 26 for further details. Source data are provided as a Source Data file. Created with Biorender.com. SUVR standardized uptake value ratio.

both associated with Mic-ADPRS in preclinical AD—as intermediate biomarker read-outs in future AD prevention trials modulating microglia.

Our approach significantly extends and improves previously published methods to determine cell type specificity in AD dementia heritability[5–11] in the following aspects. First, we derive individual-level PRS to not only assess cell-type-specific genetic contributions to the final outcome of AD dementia but also examine the impact of cell-type-specific AD genetic risk on key pathophysiologic events in AD. This is a significant advancement toward the genetic dissection of distinct AD endophenotypes in humans. Second, by combining summary statistics-driven optimization of Bayesian LD shrinkage (PRS-CS) with cell-type-specific partitioning of PRS, our approach improved statistical power while avoiding overfitting during the target dataset-based *p*-value thresholding. Third, our approach allows straightforward adjustment of other cell-type-specific ADPRS or AD endophenotypes in the analysis by simply including them as a covariate, which is another significant advantage over purely summary statistics-based computational methods. This strength enabled us to perform complex multivariate causal modeling that included multiple cell-type-ADPRSs and AD endophenotypes in the same model, thereby clarifying the role of each cell-type-specific genetic risk in AD pathophysiology.

Several limitations should be considered in interpreting our results. First, our results establish associations between the cell-type-specific

genetic risk and AD endophenotypes but do not directly identify the mechanisms of these associations. Second, our study is underpowered to detect weaker cell-type-specific ADPRS–endophenotype associations or weak statistical interactions between the PRS and age, sex, or *APOE* ε4. Third, our results may have been affected by the parameters and base datasets we chose, such as the number of genes to be considered cell-type-specific (top 10%), genomic margin near the target genes (30 kb), the snRNA dataset, and the base GWAS. To maximize comparability, we followed the convention adopted by previous studies on cell-type-specific heritability of AD dementia[9–11] (e.g., using top 10% of the cell-type-specific genes). The rationales for choosing the particular snRNA dataset[43] and the base GWAS[5] are detailed in Methods, and we showed that the association between cell-type-specific ADPRS and AD endophenotypes was robust to the genomic margins selected. Fourth, our cell-type-specific ADPRS excludes the larger *APOE* region and cannot assess the cell-type-specific impact of genes within this region. Also, our study focuses on common variants and does not consider rare variants that may have strong effects on AD endophenotypes. Fifth, our study focuses on comparing across major cell types, but it is well known that diverse cell states (subtypes) exist within each cell type, including disease-associated cell states[43,55]. Also, we could not examine AD genetic risk localizing to endothelial cells and pericytes in this study, given the small number of vascular niche cell types profiled in a typical snRNA-seq study. With increasing sample size and newer methods[56], we hope to

assess cell-state-specific AD risk targeting rarer cell states—including endothelial and pericytes subtypes—in the near future. Finally, we limited our study to participants of European ancestry, as well-powered AD GWAS summary statistics were only available from individuals of European ancestry. Our current results may not generalize to other ancestries, and well-powered AD GWAS from non-European ancestries are urgently required to address the racial and ethnic disparities in AD genomics research.

Despite these limitations, our study leveraged two well-characterized datasets to reveal robust and coherent direct associations of AD genetic risk localizing to different glial cell types with distinct disease processes in AD, including in the preclinical stage. Further, our cell-type-specific PRS can be extended to any other phenotypes beyond AD, as far as there are available GWAS summary statistics and well-characterized target datasets. Future studies combining cell-type-specific polygenic approaches with large-scale multimodal data from deeply phenotyped cohorts and model systems could enable further causal dissection of the cellular contributions to AD pathogenesis.

## Methods

### ROSMAP: participants and phenotypic characterization

The Religious Orders Study (ROS) and the Rush Memory and Aging Project (MAP) were approved by an Institutional Review Board (IRB) of Rush University Medical Center. The ROS started in 1994 and is enrolling Catholic priests, brothers, and nuns across religious communities in the United States[39]. The MAP started in 1997 and is enrolling diverse participants from northeastern Illinois[39]. Each participant signed an informed consent, Anatomic Gift Act, and Repository Consent allowing their data to be repurposed. Both studies enrolled older participants who did not have known dementia at enrollment and agreed to organ donation after death (overall autopsy rate >85%). ROS and MAP (ROSMAP) were designed for combined analyses, and the same team of investigators at Rush Alzheimer's Disease Center (RADC) performed coordinated clinical and neuropathological assessments. By the time of their death, ROSMAP participants exhibit a broad and continuous spectrum of cognitive and functional impairment (ranging from cognitively unimpaired to dementia) and neuropathology burden (ranging from no pathology to severe neurodegenerative/cerebrovascular pathology), representing the general aging population[39]. Deceased participants of European ancestry who had quality-controlled genome-wide genetic data and immunohistochemistry evaluation of either Aβ or tau burden were included in our study ($n = 1457$).

Each participant got a comprehensive annual cognitive evaluation including the following 19 tests spanning multiple cognitive domains[39]: Word List Memory/Recall/Recognition, East Boston Immediate/Delayed Recall, Logical memory immediate/delayed, Boston Naming Test, Category Fluency, reading test (10 items), Digit Span forward/backward/ordering, Judgment of Line Orientation, Standard Progressive Matrices, Symbol Digit Modalities Test, Number Comparison, Stroop Color Naming, and Stroop Word Reading. Each participant's annual global cognitive function was defined as the average z-scores from these tests (standardized to baseline measures). Longitudinal cognitive decline was captured by a random slope of global cognitive function from a linear mixed-effect model adjusting for baseline age, sex, and years of education and their time interaction terms[39,57]. Final clinical diagnoses (cognitively unimpaired, mild cognitive impairment, and dementia) were assigned by a neurologist using all available antemortem clinical data without access to the postmortem neuropathologic evaluation[58].

A comprehensive post-mortem neuropathological evaluation was performed to quantify AD pathology (amyloid-β [Aβ] plaques and tau neurofibrillary tangles)[59,60]. Immunohistochemistry was used to assess the overall Aβ and paired helical filament tau (PHFtau) across eight brain regions (hippocampus, entorhinal cortex, mid-frontal cortex, inferior temporal cortex, angular gyrus, calcarine cortex, anterior cingulate cortex, and superior frontal cortex). Quantitative Aβ and PHFtau burdens were defined as the percentage area occupied by each pathology, averaged across the eight brain regions. In addition, we also analyzed both diffuse plaques (DP) and neuritic plaques (NP)—that are thought to reflect varying decree of local neuroglial reaction to Aβ accumulation[44,45]—and neurofibrillary tangles (NFT; tau pathology) assessed with silver-stained slides from five brain regions (entorhinal cortex, hippocampus [CA1], mid temporal cortex, inferior parietal cortex, and midfrontal cortex). Then, the count from each region was scaled with the corresponding standard deviation and averaged to derive quantitative summary measures of DP, NP, and NFT. All quantitative AD pathology variables were square-rooted for further analyses, given their positively skewed distributions. "Elevated Aβ" was defined as the Consortium to Establish a Registry for Alzheimer's Disease (CERAD) neuritic plaque score of "definite" or "probable." A pathologic AD diagnosis was made using the modified National Institute on Aging-Reagan Institute criteria[61]. A diagnosis of AD with dementia (autopsy-confirmed AD dementia) was made when a participant with pathologic AD also had the final clinical diagnosis of dementia[3]. The control group was defined as individuals without pathologic AD, who were also deemed to be cognitively unimpaired (CU) per the final clinical diagnosis.

Microglial density was assessed using microscopic examination in a subset of MAP participants, with the following morphologic criteria[50]: stage 1, not activated (thin, ramified processes); stage 2, activated without macrophage-like appearance (rounded cell body >14 μm with thickened processes); stage 3, activated with macrophage-like appearance (cell body >14 μm). The proportion of activated microglia (PAM) was defined as the square root of the proportion of stage 3 microglia[50]. We used average PAM from two neocortical regions (inferior temporal and mid-frontal) that showed significant association with AD pathology in a previous study[50].

### A4 screening data: participants and phenotypic characterization

The A4 study protocol was approved by IRBs at each participating site, and all participants signed informed consent before the study procedures. The A4 study is a secondary prevention trial that enrolled CU older adults (between age 65 and 85) with evidence of cortical Aβ accumulation on PET imaging from 67 sites in the United States, Australia, Canada, and Japan[40,62]. Inclusion criteria to select CU older adults included Clinical Dementia Rating global score of 0, Mini-Mental State Examination (MMSE) score of 25 to 30, and Logical Memory Delayed Recall (LMDR) score of 6 to 18. Six thousand seven hundred sixty-three participants underwent cognitive screening, and 4486 participants who met the cognitive inclusion criteria (i.e., CU) had a screening positron emission tomography (PET) to quantify the fibrillar Aβ burden. Although only those determined to have elevated Aβ were eligible to be randomized in the A4 clinical trial, we used all available data from the screening visits for this study. Of note, while the A4 study has recently concluded[63], the longitudinal post-randomization data, which is solely in participants who had elevated Aβ and were otherwise eligible for treatment, is not yet available for general use outside of clinical trial outcome analyses as of August 2023. Participants of European ancestry with florbetapir PET and genetic data were included in our study ($n = 2921$).

The Preclinical Alzheimer Cognitive Composite (PACC)[52], a cognitive composite optimized to detect early Aβ-related cognitive changes, was calculated by averaging z-scores of the following four cognitive tests at screening: MMSE, the Free and Cued Selective Reminding Test (total recall), LMDR, and the Digit Symbol Substitution Test.

We used [18]F-florbetapir PET as a biomarker of Aβ. The florbetapir PET was acquired between 50 and 70 min after injecting florbetapir, and a mean cortical standardized uptake value ratio (SUVR) was calculated using a whole cerebellar reference. "Elevated Aβ" was defined as with florbetapir PET cortical SUVR ≥ 1.15, or 1.15 > SUVR > 1.10 and a positive visual read[62]. We used [18]F-flortaucipir (FTP) PET as a biomarker of tau. The FTP PET was acquired between 80 and 110 min after injecting FTP, and tau was quantified using FTP SUVR from the bilateral temporal composite region of interest (ROI), which includes the entorhinal cortex, parahippocampal gyrus, fusiform gyrus, and inferior temporal cortex. These are among the earliest regions to accumulate cerebral tau pathology in AD, and the tau burden in these regions is associated with cognitive decline[16,64,65]. We did not apply partial volume correction (PVC). 3D T1-weighted brain MRI was done, and the image was processed with NeuroQuant (http://www.cortechslabs.com/neuroquant) for automated segmentation and segmental volume calculation. We used bilateral hippocampal volume (HV) as a marker of neurodegeneration, adjusting for intracranial volume (ICV).

## Genetic data acquisition, processing, and study participant selection

In ROSMAP, DNA was extracted from blood or postmortem brain tissue. Codons 112 and 158 from *APOE* exon 4 were sequenced to determine *APOE* haplotypes (ε2, ε3, or ε4). Genome-wide genotyping was performed on the Affymetrix GeneChip 6.0 platform ($n = 1878$), the Illumina OmniQuad Express platform ($n = 566$), or the (Illumina) Infinium Global Screening Array ($n = 494$)[66]. Data from each genotyping platform were processed using the same quality control (QC) pipeline, using PLINK[67]: genotype call rate (SNP) > 95%, minor allele frequency (MAF) > 0.01, non-random missingness (SNP) $p < 10^{-9}$, Hardy–Weinberg equilibrium $p < 1.0 \times 10^{-6}$, genotype success rate (individual) <95%, concordant sex, and no excess heterozygosity (individual). Closely related individuals (identity-by-descent pi-hat > 0.1) were excluded. Principal components (PCs) of the genotype covariance matrix were derived using EIGENSTRAT[68], and population outliers (including all participants of non-European descent) were removed (using the default setting) to avoid confounding by population structure. This resulted in a total of $n = 2496$ participants with quality-controlled genetic data. The variants were phased using Eagle v2.4[69], and palindromic variants were removed before imputation. Imputation was performed separately for each genotyping platform using Michigan Imputation Server[70] and the Haplotype Reference Consortium (HRC) reference panel (r1.1 2016, lifted-over to GRCh38 coordinates)[71], and variants with MAF > 0.01 and imputation quality $R^2 > 0.8$ from all three platforms were retained. We merged imputed data from all three genotyping platforms, and after removing EIGENSTRAT outliers from the merged dataset, we had $n = 2385$ participants with 6,577,494 genetic variants. We included $n = 1457$ deceased participants with *APOE* genotypes and immunohistochemistry quantification of AD pathology (Aβ or tau), who were not part of the single nucleus RNA-sequencing (snRNA-seq) dataset[43] used to define cell-type-specific gene sets.

In the A4 screening data, DNA was extracted from blood. Targeted genotyping was used to derive *APOE* ε2, ε3, and ε4 haplotypes. Genome-wide genotyping was performed for 3465 consenting A4 screen participants using Illumina Global Screening Array, resulting in 700,078 genotyped variants[26]. We used the same genetic data QC pipeline as ROSMAP. We limited our analyses to non-Hispanic Whites and removed ancestry outliers identified with the genotype PCs. We used the same imputation procedures as ROSMAP, resulting in 7,269,997 variants (GRCh38) in 3025 participants. Among these participants, 2921 participants who also had florbetapir PET and *APOE* genotypes were included in our analysis.

Imputed genotype dosages from both datasets were rounded to integers (0, 1, or 2) before use in further analyses.

## Derivation of cell-type-specific ADPRS

Derivation of cell-type-specific ADPRS requires (1) base GWAS summary statistics, (2) LD reference panel, (3) cell-type-specific gene sets, and (4) a target dataset with individual-level genotype data.

(1) Base GWAS summary statistics and (2) LD reference panel: We used the summary statistics from a large genome-wide association study of AD dementia and AD dementia by proxy (parental history of AD dementia)[5]. Among multiple AD GWASs that were recently published, we chose the study by ref. 5 because it used the European Alzheimer & Dementia Biobank (EADB) and the UK Biobank (UKBB) datasets for stage I, and thus ROSMAP or A4 was not a part of its stage I summary statistics (i.e., no sample overlap). Further, the Bellenguez et al. study[5] included the largest number of cases and identified the most genome-wide significant loci among the GWASs published before the time of our analysis. We used PRS-CS[41], a Bayesian regression approach using continuous shrinkage prior, to perform effect estimate shrinkage and derive the posterior effect size of each single nucleotide polymorphism (SNP) included in the AD GWAS summary statistics (stage I of ref. 5) while minimizing information loss. We used the 1000 Genome Project Phase 3 European subset (1000 G EUR)[72] as the LD reference panel and used the "PRS-CS-auto" option to estimate the global shrinkage parameter (φ)–that reflects the sparsity of the genetic architecture–for each chromosome. From the base AD GWAS'[5] stage I summary statistics (limited to the variants assessed in >50% of the stage I participants), φ was $1.2 \times 10^{-4} \pm 1.7 \times 10^{-5}$ (calculated per each chromosome). This parameter optimization step only uses the base GWAS summary statistics and the LD reference panel, independent of the target dataset characteristics, thereby avoiding an overfitting problem.

We only used the HapMap3[73] SNPs (~1 million SNPs) included in the 1000 G EUR reference panel for a computationally tractable estimation of φ from the AD GWAS summary statistics, as described in the original implementation of PRS-CS[41]. While HapMap3 is a relatively small reference panel, it effectively captures heritability attributable to common haplotypes when used in Bayesian genetic effect size shrinkage methods: in a previous study, PRS-CS has outperformed other PRS approaches that used the full 1000 G EUR panel (>4 million SNPs), and the PRS-CS model performance only slightly improved even when a denser reference panel was used[41].

(3) Cell-type-specific gene sets: We used published snRNA-seq data of 34,987 cells (nuclei) from the prefrontal cortex of $n = 24$ control participants from ROSMAP (no to very little pathology at autopsy)[43] to derive cell-type-specific gene sets. At the time of the analyses, this dataset was one of the largest snRNA datasets published from older adults with no to little pathology. These participants were excluded from our study to prevent reserve causation. We used the cell type annotation from the snRNA-seq study reported this data[43], and included six major brain cell types in our analyses: excitatory neurons (Ex), inhibitory neurons (In), astrocytes (Ast), microglia (Mic), oligodendrocytes (Oli), and oligodendrocyte precursor cells (Opc). Similar to the previous study[43], we excluded endothelial cells and pericytes from analyses as only a small number of cells were profiled from these cell types. We adapted a previously published method[10,11] to derive the top decile of the genes specifically expressed in each cell type. First, we identified 13,438 autosomal genes expressed in >1% of cells from one or more cell types. We excluded genes within the larger *APOE* region (*APOE* ± 1 Mb: GRCh38 19:43,905,781–45,909,393), as *APOE* ε4 has a disproportionately large effect size that would dwarf the effects of other common variants. Second, a gene expression specificity metric ("Sg") was calculated for gene $i$ in cell type $j$, by dividing the average expression of gene $i$ in cell type $j$ ($E_{ij}$) by the summation of $E_{ij}$ across all 6 cell types (i.e., $Sg_{ij} = E_{ij}/(\sum_j E_{ij})$). Third, we rank-ordered Sg

of genes expressed in each cell type and defined genes within the top decile (*n* = 1343) as cell-type-specific genes.

(4) PRS calculation in target datasets: We used PLINK 1.90[67,74] to calculate PRS. All PRSs excluded the larger *APOE* region. Cell-type-specific ADPRSs were computed using the subset of the variants located within each cell-type-specific gene set ±30 kb (per GRCh38 reference coordinates). We chose a 30 kb margin upstream and downstream to include most cis-regulatory variants of a given gene and to allow for mapping errors due to the LD structure[75,76]. Then, to ensure that our results are robust to the choice of the genomic margin, we compared our results with the results derived from PRSs using a 10 kb margin or a 100 kb margin. We also calculated conventional ("All") ADPRS for the ROSMAP and A4 participants to compare with cell-type-specific ADPRSs. We also derived cell-type-specific ADPRS with PRSet[13], a recently published alternative method, for benchmarking. We used the default setting of PRSet as recommended for pathway enrichment analyses (e.g., no *p*-value thresholding)[13], and used the same cell-type-specific gene sets as detailed above in "(3) Cell-type-specific gene sets." Each PRS was standardized before further analyses.

## Statistics & reproducibility

We have included all parent study (ROSMAP, A4) participants with non-missing values to maximize sample size; no statistical method was used to predetermine sample size. Participants with missing values were excluded from each analysis, and we indicated the number of participants for each analysis. This study only uses observational data (ROSMAP and A4 pre-randomization data), and thus randomization and blinding were not applicable. We have adjusted sex in all our analyses, and we have also tested sex interaction to examine for sex differences in the reported effects.

All statistical analyses were done with R version 4.2 (https://cran.r-project.org/). We used the UpSetR package[77] to visualize the membership of cell-type-specific genes. Correlations between cell-type-specific PRSs were examined with Pearson's correlation and were summarized with a heatmap colored by $R^2$.

In ROSMAP, we tested the association of cell-type-specific ADPRS with seven phenotypes: AD with dementia (AD dementia, binary), Aβ (continuous), diffuse plaque (DP), neuritic plaque (NP), PHFtau (continuous), NFT (continuous), and cognitive decline (CogDec, continuous). Models with AD dementia as an outcome used logistic regression controlling for *APOE* ε4 dosage (0, 1, or 2), *APOE* ε2 dosage (0, 1, or 2), age at death, sex, years of education, genotyping platform, and the first three genotype principal components (PC1-3). Models with neuropathology as an outcome used linear regression controlling for *APOE* ε4 dosage, *APOE* ε2 dosage, age at death, sex, genotyping platform, and PC1-3. Models with CogDec as an outcome used linear regression controlling for *APOE* ε4 dosage, *APOE* ε2 dosage, genotyping platform, and PC1-3; Age, sex, and years of education were already accounted for when deriving the CogDec variable from the longitudinal cognitive data. We also assess the association of Mic-ADPRS with PAM, adjusting for *APOE* ε4 dosage, *APOE* ε2 dosage, age at death, sex, genotyping platform, and PC1-3.

In A4, we tested the association of cell-type-specific ADPRS with four phenotypes: screening neocortical Aβ (continuous), baseline temporal tau (continuous), baseline HV (continuous), and screening PACC (continuous). Models with Aβ or tau as an outcome used linear regression controlling for *APOE* ε4 dosage, *APOE* ε2 dosage, age, sex, and PC1-3. Models with HV as an outcome used linear regression controlling for *APOE* ε4 dosage, *APOE* ε2 dosage, age, sex, intracranial volume (ICV), and PC1-3. Models with PACC as an outcome used linear regression controlling for *APOE* ε4 dosage, *APOE* ε2 dosage, age, sex, years of education, and PC1-3.

We calculated the false discovery rate (FDR) for each dataset (separately for ROSMAP [49 analyses] and A4 [28 analyses]) and used FDR < 0.025 (=0.05/2) as the statistical significance threshold for the main discovery analyses (Figs. 2 and 4) given two independent datasets being used for testing. Variance explained by each cell-type-specific ADPRS was quantified by comparing adjusted $R^2$ of the linear models with and without the PRS term ($\Delta R^2$). For logistic regression models (ROSMAP, models with AD dementia outcome), we compared Nagelkerke's $R^2$ (R package "fmsb") in models with and without a PRS term. Empiric confidence intervals of $\Delta R^2$ were calculated with bootstrapping (1000 repeats, R package "boot"). Empiric one-sided *p*-values to test the hypothesis that the PRS-CS scores (used in this study) explain greater $\Delta R^2$ than the PRSet scores (previously published method[13]) were derived by calculating the proportion of bootstrapped PRSet $\Delta R^2$ greater than the actual PRS-CS $\Delta R^2$. Reported effect sizes of *APOE* ε4 and ε2 were assessed in models including All-ADPRS. Mic-independent cell-type-specific associations using the following two approaches: (1) adjusting for Mic-ADPRS, and (2) deriving cell-type-specific ADPRS excluding genes overlapping with Mic-ADPRS. To ensure that the significant results were not driven by our choice of the genomic margin (genes ±30 kb) or the number of genotype PCs we covaried (three), we also performed sensitivity analyses (1) using different genomic margins for PRS derivation (genes ±10 kb or ±100 kb) or (2) adjusting for 10 genotype PCs. For the association of Mic-ADPRS with PAM, we used a nominal *p*-value threshold of *p* < 0.05, and this was a targeted, post hoc analysis. We examined the moderating effect (statistical interaction) of age, sex, and *APOE* ε4 by examining the association between the interaction terms ([age, sex, or *APOE* ε4] × [cell-type-specific ADPRS]) with AD endophenotypes and used *p* < 0.017 (=0.05/3 potentially moderating variables) as our significance threshold.

For causal mediation analysis, we used the widely accepted sequence of AD pathophysiology progression[4,44] as our prior: DP→NP→NFT→CogDec. We used R package "mediation[78]" for causal mediation analysis using a non-parametric bootstrap option with 10,000 simulations. Then, we used R package "lavaan" for structural equation modeling (SEM). The model was fitted for individuals with non-missing data (*n* = 1392) using default lavaan settings except that we used bootstrapping (10,000 iterations), using residualized variables (i.e., the residual after regressing out *APOE* ε4 and ε2, age at death, sex, genotyping platform, and PC1-3 from a linear model). Model fit was assessed with multiple indices, including Comparative Fit Index (CFI), Tucker Lewis Index (TLI), root mean square error of approximation (RMSEA), and standardized root mean square residual (SRMR).

## Reporting summary

Further information on research design is available in the Nature Portfolio Reporting Summary linked to this article.

## Data availability

ROSMAP phenotype data (demographic, neuropathology, diagnoses, and cognitive testing data) can be requested at the RADC Resource Sharing Hub at https://www.radc.rush.edu. ROSMAP genotype data can be requested at the AD Knowledge Portal under accession code syn23446022 (https://www.synapse.org/#!Synapse:syn23446022; see https://adknowledgeportal.synapse.org/Data%20Access for data access instructions). The A4/LEARN screening (pre-randomization) data (demographic, neuroimaging, cognitive testing, and genetic data) can be requested at https://ida.loni.usc.edu/. All of the primary data used in this study are individual-level human data that require the investigators to sign a data use agreement (ROSMAP phenotype and all A4 data) or a data use certificate (ROSMAP genotype data) to ensure human subject protection; data access instructions can be found in the above URLs. We are not allowed to directly share the polygenic risk scores of each individual as they are derivatives of individual-level human subject genetic data under controlled access. We have made

the PRS-CS posterior effect sizes of AD GWAS summary statistics, cell-type-specific gene tracks (genomic ranges; each track defines the list of SNPs used for each PRS), and the list of SNPs used for each PRSet score available at the AD knowledge portal under accession code syn52750861 as open data (https://doi.org/10.7303/syn52750861). Source data are provided with this paper.

## Code availability

R codes used in this study are available at: https://github.com/YangLabADRD/CellADPRS (Yang HS, GitHub, https://doi.org/10.5281/zenodo.8475, 2023).

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

## Acknowledgements

We thank the participants and study staff of the Religious Orders Study (ROS), the Rush Memory and Aging Project (MAP), and the A4 Study. This work was funded by the United States National Institutes of Health (K23AG062750 [H.-S.Y.], P30AG010161 [D.A.B.], P30AG072975 [J.A.S.], R01AG015819 [D.A.B.], R01AG017917 [D.A.B.], R01AG063689 [R.A.S. and others], U01AG046152 [P.L.D. and D.A.B.], U01AG061356 [P.L.D. and D.A.B.], and U19AG010483 [R.A.S. and others]). ROSMAP is supported by NIH grants. The A4 Study is funded by NIH grants, Eli Lilly and Co., and several philanthropic organizations.

## Author contributions

H.-S.Y., T.G., H.K.F., P.L.D., and R.A.S. designed this study, and H.-S.Y. oversaw all aspects of this study. H.-S.Y., V.M., T.G., H.K.F., and H.-U.K. devised the cell-type-specific PRS derivation method used in this manuscript, and H.-S.Y., L.T., and D.K. calculated cell-type-specific polygenic risk scores. H.-S.Y. performed statistical analysis. A.P.S., M.P., and K.A.J. contributed to flortaucipir PET data analysis (A4 dataset). J.A.S., D.A.B., and P.L.D. acquired phenotype and genetic data for the ROSMAP (including staff supervision). R.P.M., K.A.J., and R.A.S. acquired phenotype and genetic data for the A4 study (including staff

supervision). H.-S.Y. drafted the manuscript, and all authors interpreted the data and substantially revised this manuscript.

## Competing interests

H.-S.Y. has received personal fees from Genentech, Inc., outside the submitted work. T.J.H. serves on the Scientific Advisory Board for Vivid Genomics, outside the submitted work. Eli Lilly and Co. funded the A4 Study but had no direct influence in the submitted work. All other authors declare no competing interests in this work.
