## [Peer Review File · Nature Communications]

Cell-type-specific Alzheimer's disease polygenic risk scores are associated with distinct disease processes in Alzheimer's diseaseEditorial Note: This manuscript has been previously reviewed at another journal that is not operating a transparent peer review scheme. This document only contains reviewer comments and rebuttal letters for versions considered at *Nature Communications*. Mentions of prior referee reports have been redacted.

REVIEWER COMMENTS

Reviewer #1 (Remarks to the Author):

Yang et al. present a revised study investigating Alzheimer's disease polygenic risk scores (PRSs) defined based on the cell type specificity of microglial, astrocytic, and endothelial cell types. [redacted].

However, despite these improvements, there remain several concerns with the study methodology and interpretation.

Major issues:

- One major concern is that the authors propose a new method for calculating gene-set PRSs using PRS-CS adjusted effect sizes, but there is no formal benchmarking of this new method against other alternatives. While the authors mention one analysis using PRSet to calculate microglial PRS on AD diagnosis, they do not present the results for all other associations. Additionally, the reporting of PRS results lacks comprehensiveness and does not adhere to the standard in the polygenic risk score literature. E.g. no AUC or R2 metrics are reported, to indicate the discriminatory power or phenotypic variance explained by the PRS, respectively. Without these metrics, it is challenging to assess the potential of this approach for screening individuals with AD pathology.

- The authors test multiple different endophenotypes; however, these endophenotypes are related to each other. I wonder what is the correlation between all the endophenotypes and also what are their heritabilities? Some of the PRS results could be influenced by endophenotypes with minimal or non-heritable components.

- Given that previous studies have demonstrated interactions between PRS risk, age of onset, and APOE dosage (Fulton-Howard et al., doi: 10.1016/j.neurobiolaging.2020.09.014; Bellou et al., doi: 10.1016/j.neurobiolaging.2020.04.024), the authors should further investigate the interaction between these variants in more detail.

- The authors rely on two cross-sectional studies, one post-mortem (ROSMAP) and one preclinical (A4), making it challenging to determine "how and when AD genetic risk related to specific cell types contributes to disease processes." Although the authors perform causal modelling analyses to map mic- and ast-ADPRS to distinct events in the AD pathophysiologic cascade, they acknowledge that their model, derived using post-mortem cross-sectional data, cannot establish any causal relationships. Based on the A4 cohort website, "The A4 Study lasts for four and a half years, and participants will be required to visit the clinical research site once a month". Why not performing causal modelling on this longitudinal dataset?

Minor comments:

- The use of only three principal components as covariates is a concern. Typically, a larger number of PCs are used to account for population stratification.
- There is a typo on page 7, line 19: "celluar."
- There is a typo on page 10, line 10: "Fig 3a should be Fig 4."
- The ordering of the legend on page 35 is incorrect.
- On page 10, line 13, the authors state: "This type of association resembled the cell-type specific ADPRS-NP association in ROSMAP." It is unclear how these associations resemble each other and why the other associations do not. Further clarification is needed.

REVIEWER COMMENTS

Reviewer #1 (Remarks to the Author):

Yang et al. present a revised study investigating Alzheimer's disease polygenic risk scores (PRSs) defined based on the cell type specificity of microglial, astrocytic, and endothelial cell types. [redacted]

However, despite these improvements, there remain several concerns with the study methodology and interpretation.

Response: We thank the reviewer for acknowledging the significant improvement in our manuscript and bringing up important issues to be addressed. As detailed below, we have addressed all points raised by the reviewer and believe this has further improved our manuscript.

Major issues:

- One major concern is that the authors propose a new method for calculating gene-set PRSs using PRS-CS adjusted effect sizes, but there is no formal benchmarking of this new method against other alternatives. While the authors mention one analysis using PRSet to calculate microglial PRS on AD diagnosis, they do not present the results for all other associations. Additionally, the reporting of PRS results lacks comprehensiveness and does not adhere to the standard in the polygenic risk score literature. E.g. no AUC or R2 metrics are reported, to indicate the discriminatory power or phenotypic variance explained by the PRS, respectively. Without these metrics, it is challenging to assess the potential of this approach for screening individuals with AD pathology.

Response: We thank the reviewer for this comment. In our revised manuscript, we have added R2 metrics for each PRS in supplementary tables (Supplementary Tables 4-9, 18, 22, 25, and 26). Further, we benchmarked our results against PRSet for all trait analyses (Supplementary Tables 15 and 29, Supplementary Fig. 1-2).

“We also benchmarked our results against the PRSet¹³: when PRSet was used to derive PRS, many associations were no longer significant ($p > 0.05$), and all but one association (Ex-ADPRS – tau) appeared weaker (lower ΔR^2 ; **Supplementary Table 15** and **Supplementary Fig.1**), suggesting that our PRS-CS-based approach provides an increased statistical power.” (ROSMAP results; Page 8 lines 19-22)

“None of the trait – cell-type-specific ADPRS associations, except the Mic-ADPRS – A β pair, was significant when PRSet was used ($p > 0.05$; **Supplementary Table 29**, **Supplementary Fig.2**).” (A4 results; Page 11 lines 16-18)

We note that most of the cell-type-specific ADPRSs showed small R^2 values (ΔR^2) (mostly < 0.02). This is not unexpected, as AD has a low SNP heritability after excluding the *APOE* locus (e.g., PMID 32968074: AD dementia $R^2 \sim 0.01$ for PRS excluding chromosome 19 [*APOE*]). Given the limited SNP heritability of AD, ADPRS of any form (after excluding the *APOE* locus) would have a limited predictive value for clinical trial screening or clinical prognostication. Nonetheless, we are still very excited about potential applications of cell-type-specific ADPRS, as these scores can be leveraged to clarify how cell-type-specific AD genetic risk contributes to detailed AD endophenotypes, as we show in our study. Further, cell-type-specific ADPRS may guide sample selections in future mechanistic studies using human-derived cell lines. We summarized the implications of our approach and findings as follows, clearly stating both limitations and strengths.

“It is important to note that the association of each cell-type-specific ADPRS with AD endophenotypes was much weaker than that of *APOE* loci, and the current cell-type-specific ADPRS is unlikely to be a useful tool for clinical trial screening or disease risk stratification. Nonetheless, our study demonstrates that cell-type-specific PRS can be used to gain deeper pathophysiologic insights from well-characterized cohorts and guide future mechanistic and clinical-translational studies. For example, cell-type-specific PRS could be leveraged for a genetically guided sampling of induced pluripotent stem cell (iPSC) lines for specific cell type differentiation, or it can be used for cell-type-specific pharmacogenomic studies of anti-A β immunotherapies. Further, our study provides genetic support to consider in vivo A β and tau PET—both associated with Mic-ADPRS in preclinical AD—as intermediate biomarker read-outs in future AD prevention trials modulating microglia.” (Page 12 lines 15-24)

- The authors test multiple different endophenotypes; however, these endophenotypes are related to each other. I wonder what is the correlation between all the endophenotypes and also what are their heritabilities? Some of the PRS results could be influenced by endophenotypes with minimal or non-heritable components.

Response: We are grateful for this insightful suggestion. We agree that the (1) correlation between endophenotypes and (2) heritability of each endophenotype are important factors to consider in our analyses.

(1) Correlation between endophenotypes

The endophenotypes tested are correlated with each other, as displayed in the following heatmaps (Pearson’s R, unadjusted correlations; left, ROSMAP; right, A4). (Previous work has suggested that the levels of A β and tau are correlated in preclinical AD, while the relationship of A β with HV and PACC is expected to be weaker at this asymptomatic stage of AD.)

While we agree that the correlation between traits is an important factor that may affect PRS analyses, we observed different cell-type-specific ADPRS associations even between closely correlated AD endophenotypes (e.g., DP and NP), demonstrating that our cell-type-specific ADPRSs can capture distinct biological signals between closely related traits.

“DP is an amorphous aggregation of A β with minimal cellular reaction representing early-stage A β plaque, while NP contains a dense core with surrounding neuroglial reaction including dystrophic neurites, activated microglia, and reactive astrocytes^{44,45}. Although DP and NP burden are highly correlated (Pearson’s $r=0.69$ in our study), only NP was associated with multiple glial ADPRS, supporting that the observed cell-type-specific ADPRS – trait association was not driven by the correlation between AD endophenotypes.” (Page 7 lines 7-12)

As the relationship among AD endophenotypes has already been extensively published from both ROSMAP and A4 (e.g., PMID 9865057, 37458272), we did not include this heatmap in our manuscript.

(2) Heritability of endophenotypes

The heritability of AD endophenotypes is not well quantified, as datasets with post-mortem or advanced neuroimaging traits are underpowered for robust heritability calculations. Nonetheless, we thank the reviewer for bringing up this point, as we suspect that the largely null results for hippocampal volume and PACC in A4 might have been driven by no to low AD-related heritability in these traits among cognitively unimpaired older adults who screened for the A4 study. (i.e., AD genetic variants would have had very little effect on later-stage consequences such as hippocampal atrophy and cognitive dysfunction in cognitively unimpaired individuals.). We discussed this point as a potential limitation, as follows.

“The A4 study is likely underpowered to detect the impact of cell-type-specific AD genetic risk on neurodegeneration or cognitive impairment because all participants in the A4 screening dataset were CU without extensive AD-related neurodegeneration or cognitive decline.” (Page 11 lines 9-11)

“Second, our study is underpowered to detect weaker cell-type-specific ADPRS–endophenotype associations, especially in the CU older adults in the A4 dataset with limited AD-related neurodegeneration and cognitive changes.” (Page 13 lines 14-16)

- Given that previous studies have demonstrated interactions between PRS risk, age of onset, and APOE dosage (Fulton-Howard et al., doi: 10.1016/j.neurobiolaging.2020.09.014; Bellou et al., doi: 10.1016/j.neurobiolaging.2020.04.024), the authors should further investigate the interaction between these variants in more detail.

Response: We have examined the interaction between PRS and age, sex, and *APOE* ϵ 4 dosage, but we apologize that we did not make this clearer. Now we note that the statistical interaction analyses between these terms ruled out effect moderation.

“None of the observed trait – cell-type-specific ADPRS associations showed significant statistical interactions with age, sex, or *APOE* ϵ 4 dosage (=no effect moderation).”
(Results (ROSMAP): page 8 lines 22-24)

“None of the observed trait – cell-type-specific ADPRS associations showed significant statistical interactions with age, sex, or *APOE* ϵ 4 dosage (=no effect moderation).”
(Results (A4): page 11 lines 18-20)

- The authors rely on two cross-sectional studies, one post-mortem (ROSMAP) and one preclinical (A4), making it challenging to determine "how and when AD genetic risk related to specific cell types contributes to disease processes." Although the authors perform causal modelling analyses to map micro and ast-ADPRS to distinct events in the AD pathophysiologic cascade, they acknowledge that their model, derived using post-mortem cross-sectional data, cannot establish any causal relationships. Based on the A4 cohort website, "The A4 Study lasts for four and a half years, and participants will be required to visit the clinical research site once a month". Why not performing causal modelling on this longitudinal dataset?

Response: While the A4 study has recently concluded, the longitudinal data is still being investigated for clinical trial outcome analyses and is not yet available for other projects, including all genetic analyses. The longitudinal data will become available by June 2024, but not before the revision deadline of this paper. We believe this cross-sectional study of the screening data is an important standalone manuscript, as the longitudinal data from A4 will only include participants who had elevated A β and were eligible for randomization in the study. We plan to perform a follow-up study utilizing the longitudinal A4 data to ask

specific questions within that cohort (that only includes individuals with elevated A β). We added the clarification as follows.

“Of note, while the A4 study has recently concluded⁶³, the longitudinal post-randomization data, which is solely in participants who had elevated A β and were otherwise eligible for treatment, is not yet available for general use outside of clinical trial outcome analyses as of August 2023.” (Page 17 lines 5-8)

Minor comments:

- *The use of only three principal components as covariates is a concern. Typically, a larger number of PCs are used to account for population stratification.*

Response: We thank the reviewer for bringing up this important point. Given the relatively modest sample sizes of the deeply phenotyped cohorts used in this study (ROSMAP, A4), most prior genetic association studies utilizing these datasets have used three genotype PCs (e.g., White CC et al., *PLOS Med* 2017; Yang HS et al., *Neuron* 2020; Raghavan NS et al., *JAMA Neurol* 2020). However, we agree with the reviewer that it is important to thoroughly rule out effects driven by population stratification, and we have also included the results using 10 PCs for the significant associations found in ROSMAP or A4 at FDR<0.025 (Supplementary Tables 14 and 28), observing very similar results throughout.

- *There is a typo on page 7, line 19: "celluar."*

Response: We apologize for the typo, and it was corrected.

- *There is a typo on page 10, line 10: "Fig 3a should be Fig 4."*

Response: We apologize for the typo, and it was corrected.

- *The ordering of the legend on page 35 is incorrect.*

Response: We apologize for the error, and we corrected the labels in the legend.

- *On page 10, line 13, the authors state: "This type of association resembled the cell-type specific ADPRS-NP association in ROSMAP." It is unclear how these associations resemble each other and why the other associations do not. Further clarification is needed.*

Response: Thank you. We clarified this sentence as follows.

“These results indicate that the genetic architecture of A β measured by PET is more similar to NP (**Fig. 2d**, Ast/Mic/Oli-ADPRS associations) rather than DP (**Fig. 2e**, only Ast-ADPRS association); this is likely because the PET radiotracers for A β have a greater affinity to NP than DP^{53,54}.” (Page 10, lines 14-17)

REVIEWERS' COMMENTS

Reviewer #1 (Remarks to the Author):

Yang and colleagues present a revised manuscript looking at cell-type specific PRSs for AD, to investigate how genetic liability in specific brain cell-types may contribute to different AD endophenotypes ($A\beta$, tau, cognitive decline). This work uses an interesting approach that combines gene-set PRSs and endophenotype level data to identify new associations between AD associated GWAS signals in specific cell types and biological processes related to AD.

The authors have made significant improvements in this revised manuscript. They have included further details on the ADPRS results, provided information about the correlation of the endophenotypes studied, and clarified the limitation of using longitudinal data at the moment.

However, there are a few minor points that will improve the manuscript and that will help the reader understand and interpret the results:

- I appreciate that the authors now include the R^2 estimates. However I think those R^2 estimates should be included in the main text (instead of only in the supplementary tables). In addition, the authors should emphasize the low phenotypic variance explained by the PRS, and highlight this in the main figure, as it is key to interpret the main results and their utility.

- The authors present results comparing their pathway-specific method with a published one (PRSet). In their method, they use a PRS-CS to account for LD by shrinking the GWAS effect size estimates. In contrast, the PRSet method uses a clumping and thresholding approach in which only the lead SNP is kept in a specific LD window. The two distinct methodological approaches seem to lead to a very different number of SNPs included in each cell-type-specific ADPRS (4X higher number of SNPs for PRS-CS-based approach compared to PRSet, as per Supplementary Table 1), probably leading to the differences in the performance of the two approaches. It would be helpful if the authors (1) explain in the main text what are the key differences between the two methodologies (2) test whether their R^2 results are statistically different (the error bars in the supplementary figure seem to overlap between the two methods), and (3) explain in the main text how the method choice has affected the number of SNPs retained in each cell-type specific ADPRSs, and ultimately the observed results.

- To increase transparency and reproducibility, the authors should make publicly available the PRSs. That is, not only the number of SNPs included in each PRS (included in Supplementary Table 1), but also the list of SNPs used for each PRS (in PRSet and PRS-CS), and the effect sizes used for each method (In the case of PRS-CS this would be after shrinkage).

- If the current datasets lack power to accurately determine the genome-wide heritability of an endophenotype, how can we be confident that these analyses, which rely on a partial representation of the genome-wide risk, and are calculated on relatively small sample sizes, have enough statistical power

to predict phenotypic variance for these endophenotypes? Low power could explain why, unlike in some other studies, the authors did not find any significant interactions between PRS, age, and APOE e4 dosage. It is crucial for readers to recognize the potential challenge of low statistical power that may underlie these findings, and this emphasis should extend beyond just the A4 dataset.

REVIEWERS' COMMENTS

Reviewer #1 (Remarks to the Author):

Yang and colleagues present a revised manuscript looking at cell-type specific PRSs for AD, to investigate how genetic liability in specific brain cell-types may contribute to different AD endophenotypes (A β , tau, cognitive decline). This work uses an interesting approach that combines gene-set PRSs and endophenotype level data to identify new associations between AD associated GWAS signals in specific cell types and biological processes related to AD.

The authors have made significant improvements in this revised manuscript. They have included further details on the ADPRS results, provided information about the correlation of the endophenotypes studied, and clarified the limitation of using longitudinal data at the moment.

However, there are a few minor points that will improve the manuscript and that will help the reader understand and interpret the results:

- I appreciate that the authors now include the R2 estimates. However I think those R2 estimates should be included in the main text (instead of only in the supplementary tables). In addition, the authors should emphasize the low phenotypic variance explained by the PRS, and highlight this in the main figure, as it is key to interpret the main results and their utility.

Response: We added the R2 estimates in the main figures (Figures 2 and 4), and further clarified the small effect size as follows.

“It is important to note that each cell-type-specific ADPRS explained 3% or less of the variance in each AD endophenotype. Thus, the current cell-type-specific ADPRS is unlikely to be a useful stand-alone tool for clinical trial screening or disease risk stratification.” (Page 13, lines 15-17)

- The authors present results comparing their pathway-specific method with a published one (PRSet). In their method, they use a PRS-CS to account for LD by shrinking the GWAS effect size estimates. In contrast, the PRSet method uses a clumping and thresholding approach in which only the lead SNP is kept in a specific LD window. The two distinct methodological approaches seem to lead to a very different number of SNPs included in each cell-type-specific ADPRS (4X higher number of SNPs for PRS-CS-based approach compared to PRSet, as per Supplementary Table 1), probably leading to the differences in the performance of the two approaches. It would be helpful if the authors (1) explain in the main text what are the key differences between the two methodologies (2) test whether their R2 results are statistically different (the error bars in the supplementary figure seem to overlap between the two methods), and (3) explain in the main text how the method choice has affected the number of SNPs retained in each cell-type specific ADPRSs, and ultimately the observed results.

Response: We clarified the differences between the PRS derivation methods (PRS-CS vs. LD clumping used in PRSet) and how they affected the number of SNPs included in the PRS. (Addressing comments (1) and (3))

“PRS-CS assigns posterior effect sizes for each genetic variant based on the GWAS summary statistics and linkage disequilibrium (LD) structure and does not prune the linked SNPs. Thus, PRS-CS retains more SNPs and reduces information loss, compared to the widely used LD clumping methods that only retain one lead SNP per LD block (**Supplementary Table 1**).”
(Page 6, lines 17-21)

We have examined the statistical differences between R2 estimates by deriving empiric bootstrap p-values (defined as the proportion of the bootstrapped PRSet R2 estimates greater than the actual PRS-CS R2 estimate) and have included them in Supplementary Figures 1 and 2. We also discuss how the differences in the PRS derivation methods (PRS-CS vs. LD clumping as in PRSet) might have led to differences in observed results. (Addressing comments (2) and (3))

“We also benchmarked our results against the PRSet¹³, a previously published gene-set-based PRS approach that uses an LD clumping approach. Our PRS-CS-based cell-type-specific ADPRSs explained greater variances (ΔR^2) than PRSet scores in 14 out of 15 endophenotype-PRS associations (**Supplementary Table 15** and **Supplementary Fig.1**; 10 with empiric bootstrap p-value<0.05). Thus, our PRS-CS-based approach—that retains more cell-type-specific variants (**Supplementary Table 1**) and local genomic information—showed a superior statistical power than the existing LD-clumping-based approach.” (ROSMAP: Page 9, lines 16-22)

“Our PRS-CS-based cell-type-specific ADPRSs explained greater variances (ΔR^2) than all corresponding PRSet scores (**Supplementary Table 29**, **Supplementary Fig.2**; 3 out of 6 with empiric bootstrap p-value<0.05).” (A4: Page 12, lines 16-18)

- To increase transparency and reproducibility, the authors should make publicly available the PRSs. That is, not only the number of SNPs included in each PRS (included in Supplementary Table 1), but also the list of SNPs used for each PRS (in PRSet and PRS-CS), and the effect sizes used for each method (In the case of PRS-CS this would be after shrinkage).

Response: We have deposited our data on the Alzheimer’s Disease (AD) Knowledge Portal (<https://adknowledgeportal.synapse.org/>; Project SynID: syn52750861). This includes **(1) the PRS-CS posterior effect sizes of AD GWAS summary statistics** and **(2) cell-type-specific gene tracks (genomic ranges)**. All SNPs (with posterior effect sizes) that are included in each genomic range were used to calculate each PRS-CS-based cell-type-specific ADPRS. The PRSet uses the same genomic range, but the original AD GWAS summary statistics (Bellenguez et al., stage I; reference 5) and also performs LD pruning. We have uploaded **(3) the list of SNPs for each PRSet-based scores**. The codes used for cell-type-specific ADPRS derivation are available at <https://github.com/YangLabADRD/CellADPRS>.

Please note that public sharing of the calculated PRS is not allowed, as these are individual-level human subject genetic data that is under controlled access (<https://help.adknowledgeportal.org/apd/FAQ.2635956331.html#FAQ-HowdoItratControlledAccessdataintermsofdatareuse,sharing,andpublication?>). We have noted instructions on how to access ROSMAP and A4 genetic data in the updated Data Availability statement:

“**Data Availability:** ROSMAP phenotype data (demographic, neuropathology, diagnoses, and cognitive testing data) can be requested at the RADC Resource Sharing Hub at <https://www.radc.rush.edu>. ROSMAP genotype data can be requested at the AD Knowledge Portal under accession code syn23446022 (<https://www.synapse.org/#!Synapse:syn23446022>; see <https://adknowledgeportal.synapse.org/Data%20Access> for data access instructions). The A4/LEARN screening (pre-randomization) data (demographic, neuroimaging, cognitive testing, and genetic data) can be requested at <https://ida.loni.usc.edu/>. All of the primary data used in this study are individual-level human data that require the investigators to sign a data use agreement (ROSMAP phenotype and all A4 data) or a data use certificate (ROSMAP genotype data) to ensure human subject protection; data access instructions can be found in the above URLs. We made the PRS-CS posterior effect sizes of AD GWAS

summary statistics, cell-type-specific gene tracks (genomic ranges; each track defines the list of SNPs used for each PRS), and the list of SNPs used for each PRSet score available at the AD knowledge portal under accession code syn52750861 as open data (DOI: <https://doi.org/10.7303/syn52750861>). Source data are provided with this paper.”

- If the current datasets lack power to accurately determine the genome-wide heritability of an endophenotype, how can we be confident that these analyses, which rely on a partial representation of the genome-wide risk, and are calculated on relatively small sample sizes, have enough statistical power to predict phenotypic variance for these endophenotypes? Low power could explain why, unlike in some other studies, the authors did not find any significant interactions between PRS, age, and APOE e4 dosage. It is crucial for readers to recognize the potential challenge of low statistical power that may underlie these findings, and this emphasis should extend beyond just the A4 dataset.

Response: Thank you. We have noted this point as follows:

“Second, our study is underpowered to detect weaker cell-type-specific ADPRS–endophenotype associations or weak statistical interactions between the PRS and age, sex, or APOE ε4 .”
(Page 13, lines 14-16)